# Sleep Quality and Duration in Children That Consume Caffeine: Impact of Dose and Genetic Variation in *ADORA2A* and *CYP1A*

**DOI:** 10.3390/genes14020289

**Published:** 2023-01-22

**Authors:** Chaten D. Jessel, Ankita Narang, Rayyan Zuberi, Chad A. Bousman

**Affiliations:** 1Cumming School of Medicine, University of Calgary, Calgary, AB T2N4N1, Canada; 2Alberta Children’s Hospital Research Institute, University of Calgary, Calgary, AB T2N4N1, Canada; 3Mathison Centre for Mental Health Research & Education, Hotchkiss Brain Institute, University of Calgary, Calgary, AB T2N4N1, Canada; 4Departments of Medical Genetics, Psychiatry, Physiology & Pharmacology, and Community Health Sciences, University of Calgary, Calgary, AB T2N4N1, Canada

**Keywords:** caffeine, sleep, genotype, dose–response

## Abstract

Caffeine is the most consumed drug in the world, and it is commonly used by children. Despite being considered relatively safe, caffeine can have marked effects on sleep. Studies in adults suggest that genetic variants in the adenosine A2A receptor (*ADORA2A*, rs5751876) and cytochrome P450 1A (*CYP1A*, rs2472297, rs762551) loci are correlated with caffeine-associated sleep disturbances and caffeine intake (dose), but these associations have not been assessed in children. We examined the independent and interaction effects of daily caffeine dose and candidate variants in *ADORA2A* and *CYP1A* on the sleep quality and duration in 6112 children aged 9–10 years who used caffeine and were enrolled in the Adolescent Brain Cognitive Development (ABCD) study. We found that children with higher daily caffeine doses had lower odds of reporting > 9 h of sleep per night (OR = 0.81, 95% CI = 0.74–0.88, and *p* = 1.2 × 10^−6^). For every mg/kg/day of caffeine consumed, there was a 19% (95% CI = 12–26%) decrease in the odds of children reporting > 9 h of sleep. However, neither *ADORA2A* nor *CYP1A* genetic variants were associated with sleep quality, duration, or caffeine dose. Likewise, genotype by caffeine dose interactions were not detected. Our findings suggest that a daily caffeine dose has a clear negative correlation with sleep duration in children, but this association is not moderated by the *ADORA2A* or *CYP1A* genetic variation.

## 1. Introduction

Caffeine use by children is common and considered relatively safe [1]. However, caffeine has physiological side effects among which, its ability to disturb sleep quality and duration are the most recognized and reported in children [2,3]. Given the importance of sleep to brain development [4], recommendations for caffeine consumption in children have been published by the American Academy of Child and Adolescent Psychiatry (AACAP) [5], Health Canada [6], and the European Food Safety Authority (EFSA) [7]. According to the AACAP, Health Canada, and EFSA, children and adolescents under the age of 18 years should consume no more than 100 mg/day, 2.5 mg/kg/day, and 3.0 mg/kg/day, respectively. The AACAP further recommends that caffeine use should be completely avoided for children under the age of 12. However, a safe dose of caffeine for children has yet to be established and it is unclear whether the current recommendations are sufficient for guarding against disruptions to sleep quality and duration for all children.

Sensitivity to caffeine, like any other drug, can vary from person-to-person and may, in part, result from interindividual variations in genes involved in caffeine pharmacodynamics [8] and pharmacokinetics [9]. Genome-wide association studies have found that genetic variants in the adenosine A2A receptor (*ADORA2A*) and the cytochrome P450 1A (*CYP1A*) loci are correlated with caffeine-associated sleep disturbance [10] and caffeine metabolism [11], respectively. For *ADORA2A,* rs5751876 C homozygotes were more likely to experience sleep disturbances when consuming caffeine [10], concurring with numerous candidate gene studies [12,13,14,15,16]. For the *CYP1A* locus, carriers of the T allele in the rs2472297 promoter region variant, located upstream of *CYP1A1* and *CYP1A2,* reported increased caffeine consumption and had lower plasma caffeine concentrations compared to their C allele carrying counterparts [17,18,19]. Likewise, a meta-analysis of eight adult studies found that *CYP1A2* rs762551 A carriers (also known as *1F) had a significantly higher CYP1A2 enzyme activity, as measured by caffeine metabolism, than individuals who did not carry the *1F variant [20]. These associations have potential clinical implications for children but the evidence to date has been exclusively generated in adult populations.

In the current study, we examined the independent and interaction effects of caffeine intake and candidate variants in *ADORA2A* and *CYP1A* on the sleep quality and duration in children. We hypothesized that a higher caffeine intake would be associated with greater odds of disruptions in sleep quality and duration. We also hypothesized that *ADORA2A* rs5751876 CC, *CYP1A* rs2472297 CC, and *CYP1A2* rs762551 CC carriers would have greater odds of reporting caffeine-associated disturbances in sleep quality and duration, compared to children that carried the *ADORA2A* rs5751876 T, *CYP1A* rs2472297 T, and *CYP1A2* rs762551 A (*1F) alleles.

## 2. Materials and Methods

### 2.1. Participants

Participants were drawn from the 11,875 children enrolled in the Adolescent Brain Cognitive Development (ABCD) study [21]. The ABCD study began in 2015 and is the largest, prospective, longitudinal, multi-site project designed to study brain and cognitive development in youth, as they transition into adolescence and young adulthood, across the United States [21]. At baseline, only children aged 9–10 years old were enrolled into the ABCD study. Anonymized baseline data (release 3.0) collected from enrolled children were downloaded following approval from the National Institute of Mental Health Data Archive Data Access Committee (eRSO# 1052319). All procedures were conducted in compliance with the Declaration of Helsinki and its subsequent revisions. All participants provided written informed consent prior to participation.

For the current study, only children aged 9–10 years that reported use of caffeine in the past six months and had genomic data available were included. To mitigate the potential impact of concomitant drugs with significant effects on sleep, children that reported use of melatonin, benzodiazepines, or antihistamines in the past six months were excluded. Commonly used psychotropic medications that could impact sleep (i.e., selective serotonin reuptake inhibitors, antipsychotics, and stimulants) were not an exclusion criterion, but they were accounted for in our analysis when appropriate. We also excluded children that had a T-score greater than 65 on the Child Behavior Checklist ADHD scale [22], given the strong association between ADHD symptomatology and sleep problems [23], as well as the notion that caffeine consumption may be increased in children with ADHD [24].

### 2.2. Measures

#### 2.2.1. Caffeine Intake

Youth in the ABCD study reported caffeine use via the modified Supplemental Beverage Questions administered by a trained interviewer. The modified Supplemental Beverage Questions were previously validated and showed high correlation with urine caffeine concentrations [25]. Caffeine intake from coffee, espresso drinks, tea, soda, and energy drinks were assessed. Total caffeine intake was calculated based on the number of beverages consumed per day for the previous six months. Serving size of each type of drink was defined as 8 oz for coffee, 1 oz for espresso, 8 oz for tea, 12 oz for soda, and 8 oz for energy drinks. For children who reported consuming a drink below the defined serving size, the drink amount was coded as the fraction of the serving size. The caffeine content present in each drink was estimated using standardized values obtained from published sources (Appendix A). Caffeine intake per day (mg/day) was calculated by multiplying the number of daily servings of each type of drink by the calculated caffeine content of each drink type and summing across all reported drinks. Caffeine intake per day was then divided by the child’s weight (kg) to calculate caffeine intake per kilogram per day (mg/kg/day).

#### 2.2.2. Sleep Quality and Duration

Parents of participants administered the Sleep Disturbance Scale for Children (SDSC) [26] to assess their sleep quality and duration over the past six months. The SDSC comprised 26 questions representing the most common areas of sleep disorders in childhood and adolescence, including disorders of initiating and maintaining sleep, sleep breathing disorders, disorders of arousal, sleep-wake transition disorders, disorders of excessive somnolence, and sleep hyperhidrosis. The scale has a total score ranging from 26 to 130. Previous research has shown that scores greater than 39 correspond to the upper quartile of the normal range, and give a sensitivity of 89% and specificity of 74% for identifying children with disturbed sleep quality [26].

Sleep duration was measured by a single item on the SDSC that asked, “How many hours of sleep does your child get on most nights?” with five possible response choices: 9–11 h, 8–9 h, 7–8 h, 5–7 h, or fewer than 5 h. The American Academy of Sleep Medicine consensus recommendation suggests that children aged 6 to 12 years should sleep 9 to 12 h per 24 h for optimal health [27]. As such, we merged participants reporting 8–9 h, 7–8 h, 5–7 h, or fewer than 5 h, and compared them to those reporting 9–11 h.

### 2.3. Genotyping, Imputation, and Quality Control Procedures

ABCD participants were genotyped for 503,856 genome-wide markers using the Affymetrix SmokeScreen Array (Affymetrix, Santa Clara, CA, USA) [28,29]. PLINK v1.9 was used to perform genotyping quality control. We excluded SNPs and individuals with more than 10% missing genotype calls as well as SNPs out of Hardy–Weinberg equilibrium (*p*-value < 10^−6^) or with a minor allele frequency less than 5%. Phasing and imputation were conducted via the Michigan imputation server using the Haplotype Reference Consortium (Version r1.1 2016) reference panel with mixed ancestry option. Imputed genotypes with quality scores (r^2^) > 0.3 (N_snps_ = 32,634,646) underwent post-imputation quality control using the same procedures performed on the pre-imputed data. To assess cryptic relatedness and population stratification, linkage disequilibrium (LD) pruning was performed with a threshold of 50 SNPs, with a 5 SNPs window, and r^2^ = 0.5, followed by calculation of the identical-by-descent metric (pi_ha). Samples with pi_hat > 0.2 were excluded. To account for population stratification, principal component analysis was conducted using the smartpca module of the EIGENSOFT package [30]. Each participant’s ancestry was assigned using genomic probabilities for four biogeographical groups (African, European, East Asian, and American) available in the ABCD post stratification weights database. If a clear maximum ancestry probability (>60%) was not present for an individual, they were assigned to an ‘admixed’ ancestry group.

### 2.4. Candidate Gene and Variant Selection

Two genetic variants (*CYP1A* rs2472297 and *ADORA2A* rs5751876) identified by genome-wide association studies of caffeine metabolism [11] and caffeine sensitivity to sleep disruptions [10], as well as one variant (*CYP1A2* rs762551) identified by meta-analysis of caffeine metabolism [20], were extracted from the imputed ABCD dataset.

### 2.5. Statistical Analysis

All analyses were performed with jamovi (version 2.3.18) [31]. Receiver operating characteristic (ROC) curve analysis was conducted to estimate the area under the curve for caffeine intake, and the Youden J-index (sensitivity + specificity − 1) was calculated to identify the optimal caffeine intake threshold for undesirable sleep outcomes [32]. Binomial logistic regression models were used to examine the effect of caffeine intake, *ADORA2A* rs5751876, *CYP1A* rs2472297, and *CYP1A2* rs762551 on sleep quality (normal versus disturbed based on a SDSC cut-off of 39) and duration (9–11 h versus <9 h). Odds ratios and 95% confidence intervals were calculated and adjusted for the effects of common concomitant psychotropic medications (i.e., selective serotonin reuptake inhibitors, antipsychotics, and stimulants), gender, and population stratification (principal components 1–10). For models examining the effects of *ADORA2A* and *CYP1A* variants on sleep quality and duration, odds ratios were also adjusted for caffeine use and interaction terms between caffeine use and the included variants. Exploratory models including self-reported ethnicity were also fitted.

## 3. Results

A total of 6112 children aged 9–10 years met the criteria for inclusion in this study (Table 1). More boys than girls were included in the study cohort (54.2% versus 45.8%). The mean weight of the cohort was 38.3 kg, which is between the 85th and 95th percentile for 9–10 year-old boys and girls according to the World Health Organization’s weight-for-age reference table [33]. The study population was diverse, but most (62.7%) were of European ancestry and identified as White (48.2%) (see Appendix A for concordance between ancestry and self-reported ethnicity). The average caffeine intake was 0.58 mg/kg/day. Under half (44.9%) of the cohort reported routinely getting the recommended sleep duration and about three-quarters (76.1%) reported normal sleep quality. None of the participants reported the use of CYP1A2 inhibitors (e.g., fluvoxamine) or inducers (e.g., tobacco) included in the Flockhart Drug Interactions Table [34].

### 3.1. Effect of Caffeine on Sleep Quality and Duration

Caffeine intake was not associated with disturbed sleep quality (OR = 1.08, 95% CI = 0.99–1.18, and *p* = 0.072) but it was associated with sleep duration (OR = 0.81, 95% CI = 0.75–0.88, and *p* = 1.4 × 10^−6^) (Figure 1). The inclusion of self-reported ethnicity in these models did not meaningfully change the effect estimates. The ROC curve analysis of the caffeine intake’s ability to distinguish sleep duration above or below the recommended 9 h of sleep revealed an area under the curve of 0.57 (95% CI = 0.56–0.59, *p* < 0.001). A caffeine intake threshold of 0.10 mg/kg/day was identified as the optimal threshold (Youden J-index = 0.11) for differentiating children who did from those that did not report > 9 h of sleep per night (Table 2).

### 3.2. Effect of CYP1A and ADORA2A Genotypes on Caffeine Intake

Genotype frequencies for *CYP1A* rs2472297 and *ADORA2A* rs5751876 in the current study and those reported in the five 1000 Genomes Project populations [35] are shown in Table 3. As expected, the genotype frequencies were most closely aligned with those reported in European populations. Neither rs2472297 (*p* = 0.509), rs762551 (*p* = 0.864), nor rs5751876 (*p* = 0.159) were associated with caffeine intake (Appendix A).

### 3.3. Effect of CYP1A and ADORA2A Genotypes on Sleep Quality and Duration

Neither rs2472297, rs762551, nor rs5751876 were associated with disturbed sleep quality (Figure 2) or sleep duration (Figure 3). Likewise, genotype by caffeine intake interactions were not detected (Appendix A).

## 4. Discussion

Our findings showed the mean caffeine intake among children in this study was 0.58 mg/kg/day, which is equivalent to 22 mg or approximately 7 ounces (207 mL) of cola (e.g., Pepsi) per day for the average 38 kg child in the ABCD cohort. This average daily dose is below the recommended thresholds published by Health Canada (2.5 mg/kg/day) [6] and the EFSA (3.0 mg/kg/day) [7], but it is higher than previously reported daily intakes [36]. In a representative survey of US children aged 6–11 from 1994 to 1998 [36], mean caffeine intake estimates of 0.40 mg/kg/day were reported, supporting the notion that caffeine intake trends over time have increased in children [1].

In partial support of our hypothesis, children with a higher caffeine intake had lower odds of reporting > 9 h of sleep per night. For every mg/kg/day of caffeine consumed, there was a 19% (95% CI = 12–26%) decrease in the odds of children reporting > 9 h of sleep, in concordance with the findings in adults [37]. We further explored the sensitivity and specificity of various caffeine intake thresholds to determine whether the current recommended intake thresholds set by Health Canada (2.5 mg/kg/day) and the EFSA (3.0 mg/kg/day) were sufficient for guarding against a lower than recommend sleep duration. Our results suggested that the current recommended thresholds have a limited ability to distinguish children above or below the recommended 9 h of sleep per night. Only 1.5% and 2.4% of the cohort consumed caffeine above the Health Canada and EFSA recommended thresholds, respectively, but more than half (55.1%) of the cohort reported getting fewer than the recommended 9 h of sleep per night. Our findings suggest a significantly lower threshold of 0.10 mg/kg/day would be the most appropriate for children aged 9–10. However, the sensitivity (0.66) and specificity (0.45) of this threshold was modest, and may have limited clinical utility as indicated by a Youden J-index less than 0.50 [32].

Our findings did not support an association between caffeine intake and disturbed sleep quality. The absence of an association could be attributed to the relatively low mean caffeine intake in the cohort, the sensitivity of the SDSC measure to detect caffeine-induced disturbed sleep quality, or the presence of moderating factors in which only certain subgroups are susceptible to caffeine-induced disturbed sleep quality. This latter explanation is supported by previous work in the full ABCD cohort that reported a modest association (Spearman’s rho = 0.06) between caffeine intake and SDSC measured sleep disturbance [38]. Unlike this previous study, we excluded children that were presumed to have a higher propensity for sleep disturbance (i.e., those with high ADHD symptomatology and those taking melatonin or antihistamines), which may explain why we were unable to detect a similar association between caffeine and disturbed sleep quality. Regardless, the association between caffeine intake and disturbed sleep quality measured by the SDSC, if present, appears modest, and may have minimal clinical significance.

Our results also did not support an association between genetic variation in *ADORA2A* or *CYP1A* and sleep quality or duration. This contrasts with several adult studies that have reported that *ADORA2A* rs5751876 C homozygotes were more likely to experience sleep disturbances when consuming caffeine [10,12,13,14,15,16]. However, these adult studies included participants with significantly higher mean daily intakes of caffeine (range: 2.1 mg/kg/day–4.8 mg/kg/day) than was reported in the current study (0.58 mg/kg/day), suggesting that the moderating effect of *ADORA2A* rs5751876 on caffeine-induced sleep phenotypes may be dose-dependent. Unlike *ADORA2A*, the rs2472297 promoter region variant in the *CYP1A* locus has not been previously associated with caffeine-induced sleep phenotypes, but adult T allele carriers at this locus have been shown to consume greater amounts of caffeine and have lower caffeine plasma concentrations [17,18,19]. We did not have access to plasma concentrations and did not detect a difference in caffeine consumption by the *CYP1A* genotype, nor did genotype correlate with sleep duration or quality. The failure to detect an association with consumption could be a result of a gatekeeping effect by which parents of the participating children limit their access to caffeine, inhibiting consumption among children that might otherwise consume more caffeine. In addition, the developmental trajectory or ontogeny of CYP1A may explain the lack of association in the current study. In early childhood, CYP1A2 is expressed at 1.5 times that of adults before regressing to adult levels in early adolescence [39]. The precise level of CYP1A2 expression in children aged 9–10 is uncertain, with modeling studies suggesting that expression could be increased or equivalent to adult levels [39,40]. If CYP1A2 expression is higher among children aged 9–10, it is possible that the increased enzyme activity associated with the *CYP1A* rs2472297 T allele may have a lesser impact on caffeine plasma concentrations, consumption, and sleep phenotypes. Future works exploring whether the *CYP1A* rs2472297 variant is associated with caffeine consumption and sleep phenotypes in an age-dependent manner are warranted.

Our findings should be interpreted in the context of several caveats. First, caffeine intake was collected using a validated approach but was reliant on the recall of caffeine consumed per day for the previous six months. Second, the time of day at which caffeine was typically consumed by participating children was not collected and could modify the magnitude of caffeine’s effects on sleep phenotypes. Third, caffeine consumption was based exclusively on the intake of caffeinated beverages. The consumption of other sources of caffeine (e.g., chocolate) was not included, and as such, our caffeine consumption was likely underestimated in this study. Fourth, our results are based on cross-sectional data, prohibiting temporal relationships being established between caffeine intake and sleep duration or quality. Fifth, the causes of caffeine-associated sleep disturbances are multifactorial. We did not examine several lifestyle (e.g., physical activity, diet), medical (e.g., health conditions), psychological (e.g., trauma), and familial (e.g., family history of psychiatric conditions) factors that may impact caffeine use and sleep disturbances. The future exploration of these factors is warranted. Finally, sleep phenotypes were measured subjectively via self-report. Objective measures of sleep duration and quality via wearable devices should be considered in future studies.

In summary, the current study represents the largest examination of caffeine’s association with sleep duration and quality in children. The findings show a clear association between increased caffeine intake and reduced sleep duration and suggest that the current recommended caffeine thresholds for children may be inadequate to ensure that they obtain the recommended hours of sleep. Despite robust findings in adults that suggest *CYP1A* and *ADORA2A* genetic variations moderate caffeine intake and caffeine-induced sleep disturbance, the current study could not replicate these findings in children. As such, additional work to identify the appropriate markers of caffeine sensitivity in children is warranted.

## Figures and Tables

**Figure 1 genes-14-00289-f001:**
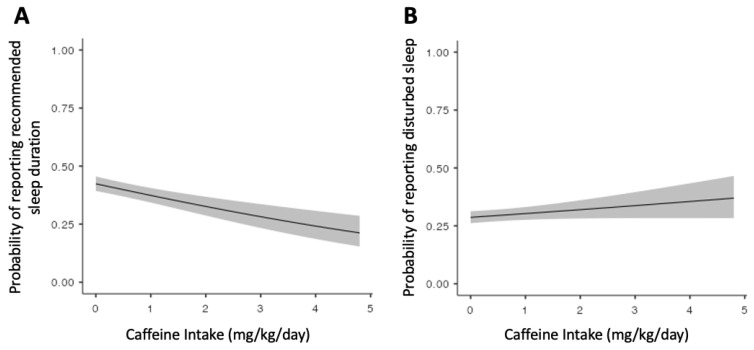
Predicted probability of reporting (**A**) recommended sleep duration (>9 h) and (**B**) disturbed sleep by caffeine intake (mg/kg/day) adjusted for psychotropic use (i.e., SSRIs, antipsychotics, and stimulants), gender, and population stratification (principal components 1–10). Shading represents 95% confidence intervals.

**Figure 2 genes-14-00289-f002:**
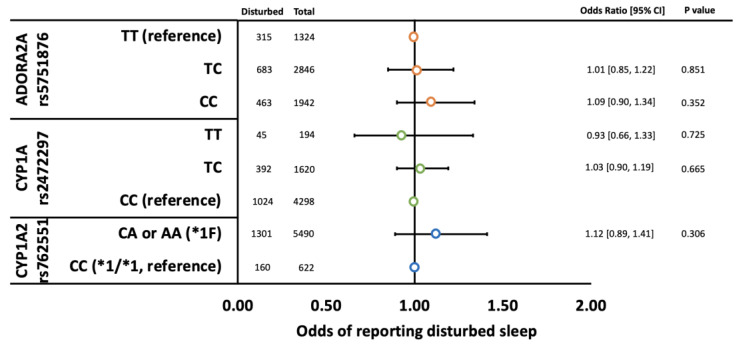
Odds of children reporting disturbed sleep by *ADORA2A* rs5751876 (orange circles), *CYP1A* rs2472297 (green circles), and *CYP1A2* rs762551 (blue circles) genotypes adjusted for psychotropic use (i.e., SSRIs, antipsychotics, and stimulants), gender, caffeine use (mg/kg/day), and population stratification (principal components 1–10). Error bars represent 95% confidence intervals.

**Figure 3 genes-14-00289-f003:**
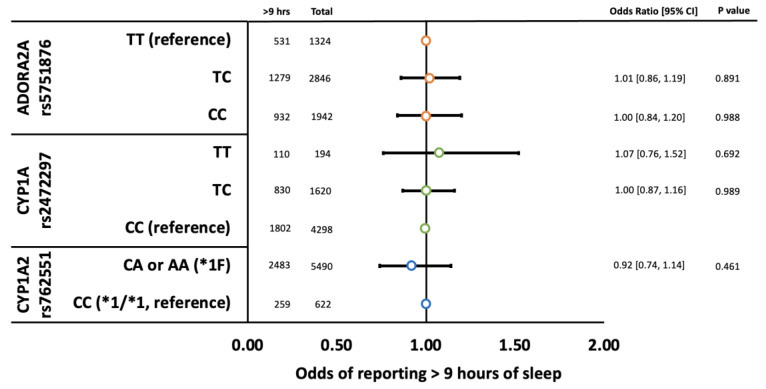
Odds of children reporting > 9 h of sleep by *ADORA2A* rs5751876 (orange circles), *CYP1A* rs2472297 (green circles), and *CYP1A2* rs762551 (blue circles) genotypes adjusted for psychotropic use (i.e., SSRIs, antipsychotics, and stimulants), gender, caffeine use (mg/kg/day), and population stratification (principal components 1–10). Error bars represent 95% confidence intervals.

**Table 1 genes-14-00289-t001:** Participant characteristics (*n* = 6112).

Characteristic	
Gender, % (N) girl	45.8 (2801)
Ancestry, % (N)	
European	62.7 (3835)
African	14.9 (914)
East Asian	0.9 (60)
American	0.8 (52)
Admixed	13.2 (810)
Not available	7.2 (441)
Ethnicity (self-reported), % N	
White	48.2 (2948)
Black	14.7 (897)
Asian	1.4 (83)
Hispanic	19.1 (1168)
Other	9.4 (574)
Not available	7.2 (442)
Weight, mean (sd) kg	38.3 (11.1)
Concomitant psychotropic use, % (N)	
SSRI ^1^	0.8 (47)
Antipsychotic ^2^	0.3 (17)
Stimulant ^3^	4.9 (298)
Caffeine use, mean (sd) mg/kg/day	0.58 (0.008)
Sleep Duration, % (N)	
9–11 h	44.9 (2742)
8–9 h	38.4 (2349)
7–8 h	13.4 (816)
5–7 h	3.2 (193)
<5 h	0.2 (12)
SDSC total score, mean (sd)	35.8 (0.09)
SDSC sleep quality, % (N) disturbed	23.9 (1461)

SDSC, Sleep Disturbance Scale for Children; SSRI, selective serotonin reuptake inhibitors. ^1^ Fluoxetine, citalopram, escitalopram, paroxetine, sertraline, fluvoxamine, and vortioxetine; ^2^ risperidone, aripiprazole, quetiapine, and olanzapine; ^3^ amphetmaine, dextroamphetamine, and methylphenidate, lisdexamfetamine. sd = standard deviation, N = number of participants.

**Table 2 genes-14-00289-t002:** Sensitivity and specificity at various thresholds of caffeine intake for detection of children reporting fewer than 9 h of sleep.

Caffeine Threshold (mg/kg/day)	Detection of Children Reporting < 9 h of Sleep Per Night		
	Sensitivity	Specificity	Youden J-Index *
0.01	0.99	0.01	0.00
0.05	0.79	0.30	0.09
0.10	0.66	0.45	0.11
0.25	0.41	0.68	0.09
0.50	0.26	0.82	0.08
1.00	0.14	0.91	0.05
1.50	0.08	0.95	0.02
2.00	0.05	0.97	0.01
2.50 ^a^	0.03	0.98	0.01
3.00 ^b^	0.02	0.99	0.01

^a^ = Health Canada recommended threshold; ^b^ = European Food Safety Authority recommended threshold; * J = (sensitivity + specificity − 1).

**Table 3 genes-14-00289-t003:** *CYP1A* and *ADORA2A* genotype frequencies in the current study and 1000 genomes populations.

				1000 Genomes Project				
SNP	Genotype	Current Study (*n* = 6112)	ALL (*n* = 2504)	AFR (*n* = 661)	AMR (*n* = 347)	EAS (*n* = 504)	EUR (*n* = 503)	SAS (*n* = 489)
rs2472297	C/C	70.3	88.0	97.1	83.0	100.0	62.0	93.5
(CYP1A)	C/T	26.5	10.9	2.9	16.1	0.0	33.0	6.5
	T/T	3.2	1.1	0.0	0.9	0.0	5.0	0.0

rs762551	C/C	10.2	15.2	19.8	7.5	10.9	11.5	22.7
(CYP1A2)	C/A or A/A	89.8	84.8	80.2	92.5	89.1	88.5	77.3

rs5751876	C/C	31.8	20.7	9.4	28.0	22.8	37.4	11.7
(ADORA2A)	C/T	46.6	47.0	44.9	52.4	48.8	47.1	44.0
	T/T	21.7	32.3	45.7	19.6	28.4	15.5	44.4

ALL, full 1000 Genomes Project sample; AFR, African; AMR, American; EAS, East Asian; EUR, European; SAS, South Asian.

## Data Availability

The ABCD data used in this report came from http://dx.doi.org/10.15154/1519007 (accessed on 11 May 2021).

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
