# Peer review of "Sleep Quality and Duration in Children That Consume Caffeine: Impact of Dose and Genetic Variation in ADORA2A and CYP1A"

_genes, 2023, doi:10.3390/genes14020289_

Round 1

Reviewer 1 Report

The topic of this manuscript is of high interest and findings of such a field of research potentially important for the community. 

The work is generally sound and well presented. Authors already discussed some of the main limitations, including missing information about the time of the day of caffein intake in children under study, which, in my opinion, could severely impact on the findings. 

Another concern relates to the complete lack of information (at least in the manuscript) regarding the use in children of other medications impacting on sleep, such as benzodiazepines or other psychotropic medications. Moreover, data on lifestyle (including sports and food), as well as socio-demographic variables (only child, divorced parents, economic status and so on) and on comorbid medical disorders, childhood traumas or family history for psychiatric disorders should be also tested and detailed in the manuscript. If all or some of these variables were exclusion criteria in the ABCD study, then this should be clearly reported in the manuscript. 

Another comments relate to the SNPs included in the analysis, especially for the CYP1A gene. SNP rs2472297 is not the only SNP in this gene that has been previously associated with altered metabolism of caffeine. Authors might want to include in the analyses other alleles previously suggested to be potentially implicated in caffein consumption as well, such as rs2470893, or other SNPs in CYP1A2 gene, such as rs762551. While the evidence for an involvement of the latter SNPs may be less robust in the literature, including them in the study might extend our understanding on the role of CYP1A in the interplay between genes, caffeine consumption and sleep. 

Reviewer 2 Report

This is a well-written manuscript with a decent presentation of study findings. The measurements and related methodology have been validated by previous research within the ABCD study. Examining caffeine consumption and sleep quality along with the pharmacogenomics implications in children is clinically important and fills the gap in current knowledge. A few comments for authors’ reference to further improve this work.

1. Was self-report race examined in this study and in comparisons of the genomic-derived ancestry? I wondered whether the non-genetic aspects of ancestry/race (e.g., cultural and lifestyle) could be associated with both caffeine intake and sleep habits as well as the associations examined in this study. 

2. Only two SNP were examined in this study. A couple of other variants with implications in caffeine metabolism and activities may also be interesting to explore. 

References:

https://www.pharmgkb.org/chemical/PA448710/clinicalAnnotation

3. Although the authors noted that children with reported use of melatonin or antihistamines in order to mitigate their impact on sleep, it would be interesting to explore whether the use pattern of those drugs is associated with caffeine intake and sleep outcomes. 

Minor:

4. It would be helpful to provide the rationale for only including children aged 9-10 years for audiences who are not familiar with the ABCD study. 

5. Page 5 - line 187: Please confirm Table 3 vs Table 2

Round 2

Reviewer 1 Report

Authors modified the manuscript according to referee's suggestions. 

Reviewer 2 Report

Thanks for taking the time to consider my suggestions. I do not have additional comments. It’s my pleasure to review this great work.